# Topological nodal line in superfluid ³He and the Anderson theorem

T. Kamppinen [1], J. Rysti[1], M.-M. Volard[1], G. E. Volovik[1,2] & V. B. Eltsov [1] ✉

Superconductivity and superfluidity with anisotropic pairing—such as $d$-wave in cuprates and $p$-wave in superfluid ³He—are strongly suppressed by impurities. Meanwhile, for applications, the robustness of Cooper pairs to disorder is highly desired. Recently, it has been suggested that unconventional systems become robust if the impurity scattering mixes quasiparticle states only within individual subsystems obeying the Anderson theorem that protects conventional superconductivity. Here, we experimentally verify this conjecture by measuring the temperature dependence of the energy gap in the polar phase of superfluid ³He. We show that oriented columnar non-magnetic defects do not essentially modify the energy spectrum, which has a Dirac nodal line. Although the scattering is strong, it preserves the momentum along the length of the columns and forms robust subsystems according to the conjecture. This finding may stimulate future experiments on the protection of topological superconductivity against disorder and on the nature of topological fermionic flat bands.

Very soon after discovery of superconductivity, H. Kamerlingh Onnes noticed that addition of impurities to a superconducting metal does not change its critical temperature. Explanation of this counterintuitive effect had to wait until formulation of the Bardeen−Cooper−Schrieffer theory, based on which P.W. Anderson proved his famous theorem[1]. The Anderson theorem states that non-magnetic impurities do not modify static properties of a superconductor with conventional $s$-wave pairing, including the critical temperature $T_c$ and value of the superconducting gap $\Delta(T)$. While origin of this robustness is closely linked to the time-reversal symmetry of the pairing state, a handwaving illustration is presented by the cartoon in Fig. 1b: Impurity scattering mixes quasiparticle states with different momentum $\mathbf{p}$ directions, but if $\Delta(\mathbf{p})$ = const then the "averaged" gap remains unchanged. For unconventional $d$-wave or $p$-wave systems, the Anderson theorem is generally not applicable. Here the gap is usually anisotropic and often includes nodes, that is, points or lines in momentum space where $\Delta(\mathbf{p})$ = 0. As the cartoon in Fig. 1c suggests, scattering then suppresses the gap[2,3], while the effect of disorder on the physics related to the energy nodes becomes a separate actively investigated problem[4−10].

For applications of unconventional and topological superconductors, strong suppression of $T_c$ is undesirable and mechanisms to improve robustness of Cooper pairs in such systems, in particular

through extensions of the Anderson theorem, are now under intensive study[11−16]. Here $p$-wave superfluid ³He provides an ideal platform to elucidate the effects of disorder: This system is naturally void of any impurities, while scattering centers, in the form of solid nanoparticles, can be immersed into the liquid under full experimentalists' control. In fact, nanostructured confinement of ³He became a flagship tool to engineer novel topological phases of matter[17−21]. Here we focus on the polar phase[22], where the confining matrix forms a set of nearly parallel strands, see Fig. 1. The polar phase is believed to have anisotropic gap with a Dirac nodal line in the plane perpendicular to strands, Fig. 1a, although an experimental confirmation for the presence of the node has so far been missing.

A remarkable feature of the polar phase is that its $T_c$ is suppressed only marginally compared to the critical temperature $T_{cb}$ of clean bulk ³He (Fig. 1d), even when the distance between scattering strands is just a fraction of the coherence length[22,23]. To explain this robustness, it has been suggested that the Anderson theorem can be extended to the polar phase[24], provided impurities have the form of infinitely long non-magnetic strands, which are straight and parallel to each other and the scattering of quasiparticles is fully specular (see Fig. 1e). The reason is that the polar phase represents a set of independent two-dimensional (2D) superfluids with different $p_z$. (Here $\hat{\mathbf{z}}$ is the direction along the

[1]Department of Applied Physics, Aalto University, POB 15100, FI-00076 AALTO, Finland. [2]Landau Institute for Theoretical Physics, 142432 Chernogolovka, Russia. ✉e-mail: vladimir.eltsov@aalto.fi

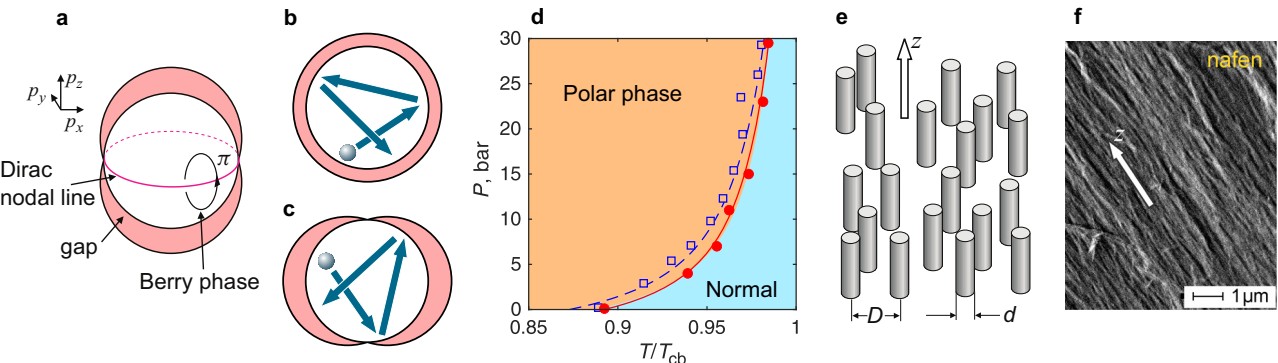

**Fig. 1 | Polar phase of superfluid ³He under nanostructured confinement. a** The topology- and symmetry-protected Dirac nodal line in the spectrum of the Bogo-liubov quasiparticles in the polar phase forms a circle in the $p_z = 0$ plane, while the superfluid energy gap is axially symmetric with respect to $\hat{z}$ axis and reaches maxima at $p_z = \pm p_F$ with $p_F$ being the Fermi momentum. The Berry phase changes by $\pi$ on a path around the nodal line, see Supplementary Note 1. **b** A cartoon illustrating the Anderson theorem: In systems with *s*-wave pairing and isotropic gap, a quasiparticle (silver ball) changes its momentum direction (blue arrows) on scattering events, but the effective gap the quasiparticle "sees" remains the same. Remarkably, similar picture applies in the *p*-wave polar phase if scattering preserves $p_z$ component of momentum. **c** In general unconventional superconductor with anisotropic gap, the Anderson theorem is not applicable and the gap is suppressed. **d** The phase diagram of superfluid ³He confined in nafen-243 nanomaterial is occupied by the polar phase, while the suppression of $T_c$ compared to the

transition $T_{cb}$ in bulk (not confined) ³He is relatively small. The circles are our measurements and squares are from ref. 22. Lines are fit to the model of ref. 26, see the text for details. **e** Idealized model of the nanostructured confinement used to engineer the polar phase: A system of randomly distributed columnar defects of diameter *d* and spacing *D*, oriented along the *z*-axis and providing specular quasi-particle scattering, which conserves the *z*-component $p_z$ of the momentum. For such model, the Anderson theorem is extended to a spin-triplet *p*-wave superfluid— the polar phase[24]. **f** A microphotograph of the nafen-243 material used for ³He confinement in this work. For this material[22] $\langle d \rangle \approx 9$ nm and $\langle D \rangle \approx 35$ nm. Unlike the perfect model in panel (**e**), the real material has orientational disorder of the $Al_2O_3$ strands, which somewhat violates the Anderson theorem and results in the small suppression of $T_c$ in the phase diagram. This imperfection is not under good control, and the $T_c$ suppression is different for the two sets of data in panel (**d**), although the two samples have the same nominal density.

strands.) For perfect columnar defects the scattering between different $p_z$ states (or 2D bands) is absent. Such 2D superfluids have the same properties as *s*-wave superconductors, including time-reversal sym-metry, the Anderson theorem is applicable and impurities do not break the Cooper pairs. Conceptually similar approach was suggested for extension of the Anderson theorem to multi-band unconventional superconductors[11–14].

Here we verify the applicability of the Anderson theorem to the polar phase with measurement of the temperature dependence of the gap $\Delta(T)$ at temperatures $T < 0.5\,T_c$. We find that $\Delta(0) - \Delta(T) \propto T^3$, which is a signature of the Dirac nodal line. Moreover, the prefactor in this cubic dependence is close to the theoretical expectation based on the BCS theory in the clean limit. Generally, disorder in nodal systems is expected to affect both the power law of the temperature dependence of various properties and the absolute magnitude of the change, see, e.g., ref. 25. Agreement of the gap measurements in the polar phase under strong columnar disorder with the clean-limit theory is a definite manifestation of the Cooper pair protection by the extended Anderson theorem. The small deviations from ideal clean-limit values in the magnitude of the gap change at low temperatures and in the critical temperature are found to have similar pressure dependence. These deviations probably originate from the real confinement providing channels for mixing between different $p_z$ subsystems, e.g., due to orientational disorder in confining strands.

## Results
### Suppression of $T_c$
In the nafen-243 material used for confinement in this work, 94% of the volume is an open space between strands. This is a relatively low level of porosity: Impurity scattering is strong with $\langle \tau\Delta/\hbar \rangle \lesssim 1$ at all pressures and temperatures, see Supplementary Fig. 1. Here $\tau$ is the quasiparticle scattering time. When ³He is confined in silica aerogels of such por-osity, superfluidity is suppressed completely and even in 98%-open silica aerogels the superfluid is suppressed at pressures below about 5 bar[3]. The reason is that more randomly distributed impurities in silica aerogels do not allow extension of the Anderson theorem to that system. On the contrary, since the original measurement in the

confinement with columnar defects[22], it is known that $T_c \gtrsim 0.9\,T_{cb}$ in nafen-243. Our measurements confirm this result, Fig. 1d.

Drastic change of $T_c$ from zero to nearly $T_{cb}$ is the result of pro-tection provided by the Anderson theorem. In ideal case of the theorem applicability, $T_c = T_{cb}$ independently of the defect density. It is inter-esting that two measurements in Fig. 1d performed on the samples of the same nominal density, demonstrate slightly different $T_c$ suppres-sion. Thus, it is not the density of defects which directly determines suppression (in accordance with the understanding provided by the Anderson theorem), but deviations of the scattering properties from ideal $p_z$-preserving. A simple model accounting for this deviation is given in ref. 26: $T_c = T_{cb}[1 - (n_s\xi_0 + \beta\xi_0^{-1})\sigma_\parallel]$. Here $n_s = 9.55 \times 10^{14}$ m$^{-2}$ is the density of nafen strands, $\xi_0$ is the pressure-dependent coherence length, $\beta$ is a parameter related to the strand diameter and $\sigma_\parallel$ is the scattering radius for a quasiparticle moving along the strand. (For ideal strand $\sigma_\parallel = 0$.) We fit two sets of data in Fig. 1d with the same $\beta$ and different $\sigma_\parallel$. The fit describes the observations reasonably well with $\beta = -0.1$ and $\sigma_\parallel = 1.7$ nm for data from ref. 22 and $\sigma_\parallel = 1.5$ nm for our data. Note that $\sigma_\parallel \ll \langle d \rangle$. Thus the orientational disorder in strands is relatively weak. Below we demonstrate that also the low-temperature measure-ments in the polar phase are described well by the clean-limit theory, while deviations are relatively small.

### Measurement of the gap
The measurement of the gap utilizes spin-orbit interaction in Cooper pairs, which have total spin 1 and orbital momentum 1. As a result of this interaction, the precession frequency of spin in nuclear magnetic resonance (NMR) experiments $\omega$ deviates from the Larmor precession frequency $\omega_L = |\gamma|H$ in the normal phase. Here $\gamma \approx -2.04 \times 10^6$ rad s$^{-1}$ T$^{-1}$ is the gyromagnetic ratio of ³He. In our measurements the magnetic field **H** is oriented along $\hat{z}$ and the frequency shift is

$$\omega(T)^2 - \omega_L^2 = \lambda_D N_0 \frac{\gamma^2}{\chi} \Delta^2(T). \tag{1}$$

Here $\lambda_D \sim 10^{-6}$ describes interaction of two magnetic dipoles in a pair and is approximately constant, $N_0$ and $\chi$ are pressure-dependent

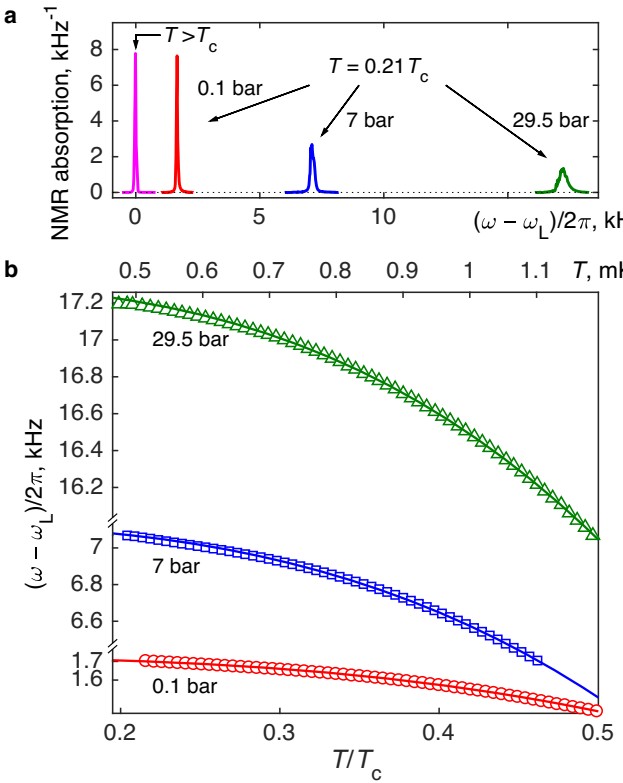

**Fig. 2 | NMR measurements of the temperature dependence of the gap in the polar phase. a** NMR spectrum of $^3$He in the polar phase ($T < T_c$) demonstrates the frequency shift from the Larmor value $\omega_L$, where the spectrum in the normal state ($T > T_c$) is located. From the temperature dependence of the shift, the temperature dependence of the gap $\Delta(T)$ can be extracted. The spectra on the plot are normalized so that the total integral of absorption is unity. **b** Temperature dependence of the frequency shift in the polar phase at the lowest temperatures at three pressures (symbols). Lines are fit of the data in the temperature range $0.3 < T/T_c < 0.5$ to Eq. (4) using $a$ and $\omega(0)$ as fitting parameters. The fitted values are shown in Fig. 3b and 4, respectively. Absolute temperatures on the upper x-axis refer to 29.5 bar pressure.

but temperature-independent density of states and magnetic susceptibility of normal $^3$He, respectively, and $\Delta(T)$ is the maximum gap in the energy spectrum of Bogoliubov quasiparticles. Examples of the NMR spectra measured in normal helium and in the polar phase at $T \approx 0.2T_c$ at different pressures are shown in Fig. 2a.

In the polar phase, the gap for arbitrary direction of momentum has the form $\Delta(T, \mathbf{p}) = \Delta(T) \cos \mu$, where $\mu$ is the angle between $\mathbf{p}$ and $\hat{\mathbf{z}}$. The gap vanishes at $\mu = \pi/2$, which gives rise to the Dirac nodal line on the equator of the Fermi surface. This nodal line is topological since the Berry phase changes by $\pi$ on a closed path around an element of the line for each of the two spin projections of fermions, see Fig. 1a and Supplementary Note 1. Due to the nodal line, the density of states in the polar phase as a function of energy $\epsilon$ is $N(\epsilon) \propto \epsilon$, which results in the cubic dependence of the free energy $F(T) - F(0) \propto T^3$ at low temperatures $T \ll T_c$. Such cubic dependence also extends to the gap amplitude[27,28]:

$$1 - \frac{\Delta(T)}{\Delta(0)} = a\frac{T^3}{T_c^3}, \quad T \ll T_c, \qquad (2)$$

where the dimensionless parameter $a$ in the BCS clean limit is

$$a \approx 8.5 \left[\frac{k_B T_c}{\Delta(0)}\right]^3, \qquad (3)$$

see Supplementary Note 2. With $\Delta(0) = 2.46T_c$ in the weak coupling approximation, the value is $a = 0.57$, see Supplementary Note 3. We remark that in the case of Weyl superfluids with point nodes (like the A phase of superfluid $^3$He), the expected temperature dependence of the gap amplitude is $(T/T_c)^4$.

From Eqs. (1) and (2) we find that

$$\frac{\omega(0) - \omega(T)}{\omega(0) - \omega_L} = 1 - \frac{\Delta^2(T)}{\Delta^2(0)} = 2a\frac{T^3}{T_c^3}, \quad T \ll T_c, \qquad (4)$$

where we assumed that $\omega - \omega_L \ll \omega_L$ and $\Delta(0) - \Delta(T) \ll \Delta(0)$. Thus the normalized frequency shift is a direct probe of the temperature dependence of the gap.

### Cubic law

We measured the NMR frequency shift in the polar phase at several pressures between 0.1 and 29.5 bar. The temperature dependence of the shift at three pressures is shown in Fig. 2b. The data fit very well with the cubic dependence of Eq. (4). The data for all pressures are presented in Supplementary Figs. 2 and 3 and the details of the fitting procedure are discussed in Supplementary Note 4. Combining uncertainties of the fit with uncertainties of the temperature calibration (see "Methods") we determine the confidence interval for the exponent in the temperature dependence of the gap as 2.9–3.2. Within this uncertainty, the power law is in full agreement with the theoretical temperature dependence of the gap in the clean limit, which comes from the Dirac nodal line.

The prefactor $a$ in the gap temperature dependence varies between 0.7 at low pressures and 0.3 at high pressures, see Fig. 3. It is comparable to its theoretical clean-limit value, although deviations from the weak-coupling value of 0.57 and systematic pressure dependence are clearly seen. Since $a \propto (T_c/\Delta(0))^3$, one contribution to this change is the increase of the $\Delta(0)/T_c$ ratio due to strong coupling effects, which in $^3$He become more important with increasing pressure[29–31]. Another possible contribution is a weak violation of the Anderson theorem due to non-ideal scattering at the strands. That is, the same effect as responsible for $T_c$ suppression. We discuss these two contributions in more details below.

We stress that beautiful agreement of the measurements with the clean-limit theory is achieved despite the fact that in our sample the impurity scattering is strong with $\tau \ll \hbar/\Delta$ in the most parts of the phase diagram. Nevertheless, as seen in Supplementary Fig. 1, at the lowest temperatures and at higher pressures $\tau\Delta/\hbar$ can reach or even exceed unity. That opens a possibility of phase transitions to different superfluid phases, not protected by the Anderson theorem. We believe that the upward deviation of the points from the fit line for 29.5 bar data at the lowest temperatures in Fig. 3a is an indication of such transition. The new phase is probably the polar-distorted A phase as observed in the less dense nafen-90 confinement as well[22]. The transition is found also at 11, 15, and 23 bar pressure with the maximum extent at 15 bar, see Supplementary Fig. 3. To avoid influence of a different phase on the polar-phase results, we limited the temperature range included in the fit in Figs. 2 and 3 to $(0.3–0.5)T_c$. See Supplementary Note 4 for details.

### Discussion

Our demonstration of the extension of the Anderson theorem to the $p$-wave system is applicable for non-magnetic scattering from impurities. This condition is fulfilled in our experiments by preplating the nafen strands with a $^4$He layer. With insufficient $^4$He coverage, the phase diagram of superfluid states in nafen changes drastically and, in particular, $T_c$ gets dramatically suppressed[23,32]. This property is in line with the well-known violation of the Anderson theorem by magnetic impurities in the $s$-wave systems.

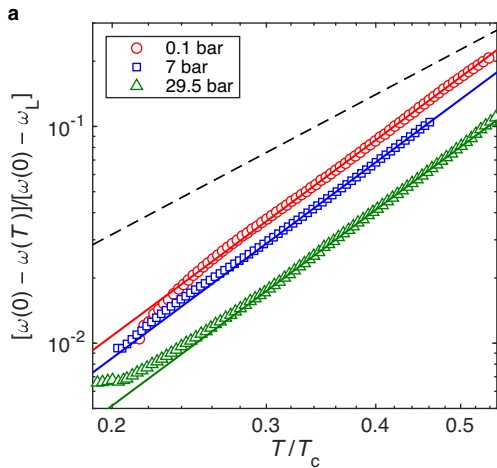

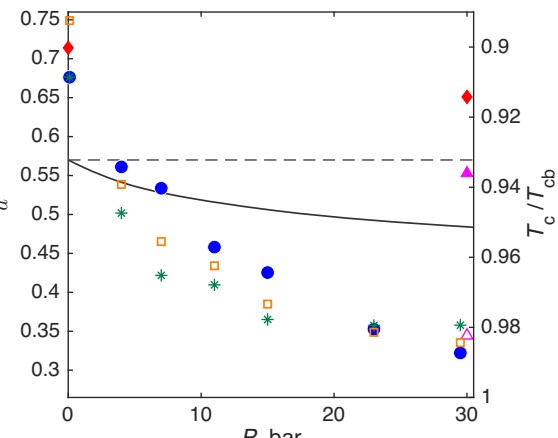

**Fig. 3 | Experimental verification of robustness of the polar phase of superfluid**
**³He with its nodal line against strong impurity scattering. a** Normalized frequency shift, measuring $1 - \Delta^2(T)/\Delta^2(0)$, shows cubic dependence on temperature. This is in agreement with Eq. (4), derived in the clean-limit BCS theory, despite strong impurity scattering imposed by the confining matrix. Cubic power law is characteristic of the presence of the nodal line. Points and solid lines (with slope 3) are the same as in Fig. 2b but replotted in appropriate coordinates. Dashed line has a slope of 2.14 suggested in one of the models of influence of disorder in nodal-line systems[25]. **b** On the left $y$ axis: The prefactor $a$ in the $T^3$-dependence of the gap as a function of pressure (circles). Its overall value agrees with calculations in the clean limit with no fitting parameters. Theoretical clean-limit value in the weak-coupling approximation is shown with the dashed line (see Supplementary Note 2). The

statistical error in $a$ is smaller than the point size (see Supplementary Fig. 3). Pressure dependence of $a$ may originate from different sources, see "Discussion". Solid line shows Eq. (3) with $\Delta(0)/T_c$ increasing with pressure owing to the strong coupling corrections as found in the B phase[29]. The stars show Eq. (3) with $\Delta(0)/T_c$ derived from the zero-temperature frequency shift in this work, see Supplementary Fig. 4. Diamonds are values calculated in a model of strongly anisotropic confinement[38], which is similar, but not identical to nafen-243 used in this work. Triangles are the 30 bar data from ref. 38 scaled with the strong-coupling corrections, see the text for details. On the right $y$ axis: Suppression of $T_c$ from Fig. 1d (squares). The scale of the axis is selected to stress similar dependence of $a$ and $T_c/T_{cb}$ on pressure.

While this channel of violation of the Anderson theorem is suppressed in the experiments, it is clear from the data, that some disorder violating the theorem remains in our system. The most likely candidate is the orientational disorder of the strands which breaks conservation of $p_z$ on scattering. The violation is seen in particular from the suppression of $T_c$ and from difference of $a$ from the weak-coupling value at the pressure of 0.1 bar where strong-coupling effects can be ignored. In case of the nodal spectrum, the disorder breaking the Anderson theorem can produce the gap in the spectrum[33]; or it can renormalize the exponent $\alpha$ in the density of states[25], $N(\epsilon) \propto \epsilon^\alpha$. It can also lead to finite density of states[34], $\alpha = 0$, and even to localization[35,36]. Different power law may come from quasiparticles living on the edges or bound to the strands of aerogel[26], or from the perturbed order parameter in the vicinity of strands[37].

In the Nersesyan et al scenario for nodal-line $d$-wave superconductors[25], the predicted power law in the density of states is $\alpha = 1/7$ with disorder instead of $\alpha = 1$ in the pure state. This corresponds to the power law $T^{2.14}$ instead of $T^3$ in the gap dependence. This scenario (which does not take into account protection by the Anderson theorem) is clearly not realized in our experiments, see the dashed line in Fig. 3a.

In the BCS-type model of Hisamitsu et al.[38], superfluid ³He is considered in the strongly anisotropic confinement which provides small, but finite change of $p_z$ momentum on quasiparticle scattering. When the parameters of the model are tuned to fit $T_c$ with nafen-243 confinement, its low-temperature predictions for the polar phase are the following: (i) $T^3$ law in the gap dependence is preserved at least until $T = 0.5T_c$; (ii) coefficient $a$ increases compared to the clean limit, see diamonds in Fig. 3b; (iii) coefficient $a$ decreases with pressure as $T_c$ suppression decreases. These predictions agree with our observations, at least qualitatively. The measured decrease of $a$ is much stronger than found in this model, but indeed has a similar pressure dependence as $T_c$ suppression, see squares in Fig. 3b.

An explanation for the enhanced pressure dependence of $a$ may come from the strong-coupling effects in superfluid ³He. These effects

results in the increase of $\Delta(0)/T_c$ with pressure and renormalization of $a$ to smaller values. The correction is well known in the B phase, but its value cannot explain our observations fully, see the solid line and the filled triangle in Fig. 3b. In principle, it is not excluded that presence of confining strands between ³He atoms interacting in a Cooper pair (since $\langle D \rangle \lesssim \xi_0$) may affect the strong-coupling effects. We can find $\Delta(0)/T_c$ from the zero-temperature frequency shift, Fig. 4, using Eq. (1), see Supplementary Fig. 4. With this correction applied with respect to 0.1 bar data, the observed overall pressure change in $a$ is reproduced, see stars and an empty triangle in Fig. 3b, although detailed pressure dependence somewhat differs from the observations. Despite the agreement, one should be careful, though, that the assumption of $\lambda_D = const$ used in this analysis, may be an oversimplification.

Overall, it is likely that orientational disorder in the strands and the strong coupling effects together explain the observed deviations in the gap temperature dependence from the prediction of the clean-limit weak-coupling theory. The detailed accounting for these effects as well as for the contribution of the strand surface-bound states which may change the temperature dependence further at the lowest temperatures, remains a task for future. Nevertheless, these effects provide just relatively small deviations from the clean-limit expectations, as confirmed by the experiment, while in the absence of protection by the Anderson theorem, the system behavior would be completely revamped.

Non-trivial symmetry and topology of the polar phase of superfluid ³He has been used to experimentally demonstrate the analog of the cosmological Alice string (the half-quantum vortex)[39]. A realization of the Kibble–Lazarides–Shafi cosmic wall bounded by strings[40] have also been observed: Those walls are formed[41] after the phase transition from the polar phase to the other phases of superfluid ³He found with the less dense confining matrix, where the scattering time is increased to $\tau \sim 2\hbar/\Delta(0)$. Our proof for existence of robust nodal line in the polar phase opens possibilities for uncovering further new physics.

Due to the presence of the node, the Landau critical velocity of superflow in the polar phase is zero. It has been recently demonstrated

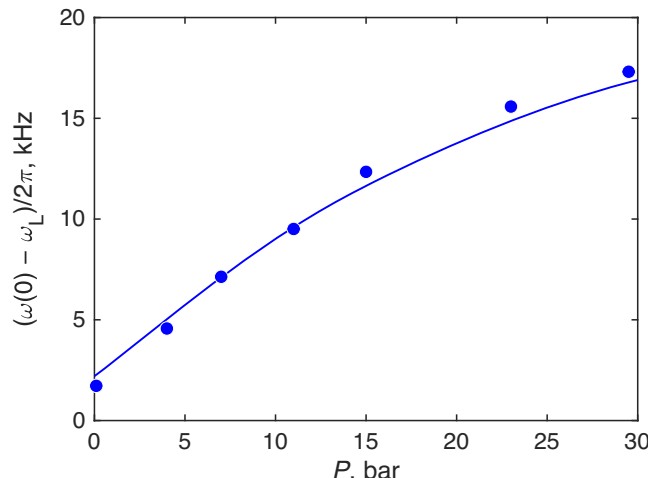

**Fig. 4 | Zero-temperature frequency shift as a function of pressure.** Frequency shift of the NMR line in the polar phase, extrapolated from fit lines in Fig. 2b to zero temperature (points). The statistical error in the shift is smaller than point size (see Supplementary Fig. 3). The line shows value from Eq. (1) with $\lambda_D = 7.5 \times 10^{-7}$ selected to fit the points. The main reason for the increase of the frequency shift with pressure is the increase of $T_c$. Additionally, the line includes increase of $\Delta(0)/T_c$ ratio with pressure due to strong-coupling corrections accepted for the B phase[29]. The data points show systematically stronger pressure dependence than the line, which may indicate larger strong-coupling corrections in the polar phase compared to the B phase, see Supplementary Fig. 4.

that the superflow in polar phase remains nevertheless stable at finite velocities due to formation of the Bogoliubov Fermi surface (BFS)[42]. Such Fermi surface, which emerges in a superconductor or a superfluid, has been suggested to exist in various systems[43–47]. In the polar phase we expect BFS to have large extent in momentum space and non-trivial topology, see Supplementary Fig. 5a. The robustness of the BFS to disorder remains an interesting question for future research, since in the presence of the superflow the Anderson theorem is not automatically applicable. The non-thermal quasiparticles located in the pockets covered by BFS reduce the average gap value. This suppression can be observed, for example, by attaching a piece of nafen with confined polar phase to a vibrating object immersed in helium or by rotating the entire sample and detecting the corresponding frequency shift in NMR, as a direct extension of the method used in the present work.

Another striking consequence of the nodal line is the presence of the topological flat band at the surface[48], normal to the direction of the confining strands, see Supplementary Fig. 5b. As a result of the topological phenomenon of bulk-boundary correspondence, the one-dimensional nodal line in bulk material generates the two-dimensional surface of zero energy states (a flat band) on the boundary. In metals and semimetals, the singular density of electronic states in the flat band leads to a linear dependence of the critical temperature $T_c$ of the superconducting transition on the strength of the pairing interaction[49]. Thus, in a flat-band system $T_c$ can be essentially higher[50] than in conventional superconductors, where $T_c$ is exponentially suppressed as a function of the pairing interaction. A particular example is provided by the superconductivity in the twisted graphene layers[51,52], while some other experiments with graphite materials demonstrate signatures of superconductivity even at room temperature[53]. In the polar phase of $^3$He, the surface flat band may give rise to a new superfluid phase on the surface with additional symmetry breaking.

This work provides an experimental evidence for the existence of the nodal line in the polar phase and demonstration of the extension of the Anderson theorem to an unconventional $p$-wave

system. Quite likely, by varying the geometry of the confinement, protection of the Anderson theorem can be extended to other superfluid phases with $p$-wave pairing, like the chiral A phase in the planar geometry[54]. The extension of the Anderson theorem has also been considered for unconventional and multi-band superconductors. Although the superconducting model systems discussed in refs. 11,12 differ significantly from the polar phase with columnar defects, the mechanism of the robustness towards disorder is the same: impurity scattering between different bands should be properly suppressed. There are also suggested scenarios in which the disorder leads to enhancement of the transition temperature due to inelastic scattering[55], and to even larger enhancement of superfluidity due to multifractality[56] of the fermion wave functions in the special arrangements of strands. At the moment we did not see any hints of the enhancement of $T_c$, but different classes of disorder can be experimentally constructed in future by varying synthesis techniques of the confinement material and its post-processing[57,58] or by nanofabrication: with different arrangements of strands, with different densities, with different fractal distributions, with different shapes of strands, with different bound states, etc. Our work opens the road to future experiments on the protection of topological superconductivity against disorder, on Bogoliubov Fermi surfaces, on fermionic topological flat bands, and on the effective metric which allows a transition to antispacetime[59].

## Methods
### Sample
The $^3$He sample is confined in a $4 \times 4 \times 4\,mm^3$ cubic container made from Stycast 1266 epoxy. The container volume is filled with a nanostructured material called nafen of 0.243 g/cm$^3$ density. The nafen consists of nearly parallel Al$_2$O$_3$ strands (Fig. 1) and provides about 94% open space within the structure. The nafen was produced by AFN Technology Ltd in Estonia. To avoid formation of paramagnetic solid $^3$He layer on the strands, which breaks the requirement of non-magnetic specular scattering, all surfaces are preplated[23] by about 2.5 monolayers of $^4$He. The $^3$He pressure is regulated between 0.1 and 29.5 bar through a filling line from a room-temperature gas handling system. The container with confined sample is connected to a larger volume of bulk $^3$He which in turn is attached to a copper nuclear demagnetization cooling stage through a heat exchanger made from sintered silver powder. The temperature is regulated by changing current in the solenoid creating the demagnetization field. Depending on pressure, the lowest temperatures reached vary between 0.19 and 0.21$T_c$ or between 200 and 450 μK. The lowest temperature is limited by the residual heat leak to the sample (about 25 pW) and the thermal resistance of the sinter enhanced by the $^4$He layer[60].

The SEM photograph of the nafen material in Fig. 1f is acquired with Jeol JSM-7100F microscope using acceleration voltage of 5 kV and working distance of 9.3 mm. The imaged sample is from the same production batch as the one used in confining superfluid $^3$He, but is a physically different piece.

### Nuclear magnetic resonance measurements
NMR spectrometer includes pick-up coils made from copper wire, which enclose the sample. The same pair of coils with the axis transverse to the static NMR field **H** is used both to excite and detect the nuclear magnetization precession. The coils are part of a tuned tank circuit, with a Q-value of 150. Signal from the pick-up coils is pre-amplified with a cold amplifier thermalized to a 4K plate and then fed for further amplification and detection using a lock-in amplifier located at room temperature.

In the measurements, the excitation is continuously applied at frequency $\omega_{rf}/2\pi = 363$ kHz, which is fixed to the resonance frequency of the tank circuit, while the magnitude of the magnetic field $H$ is swept

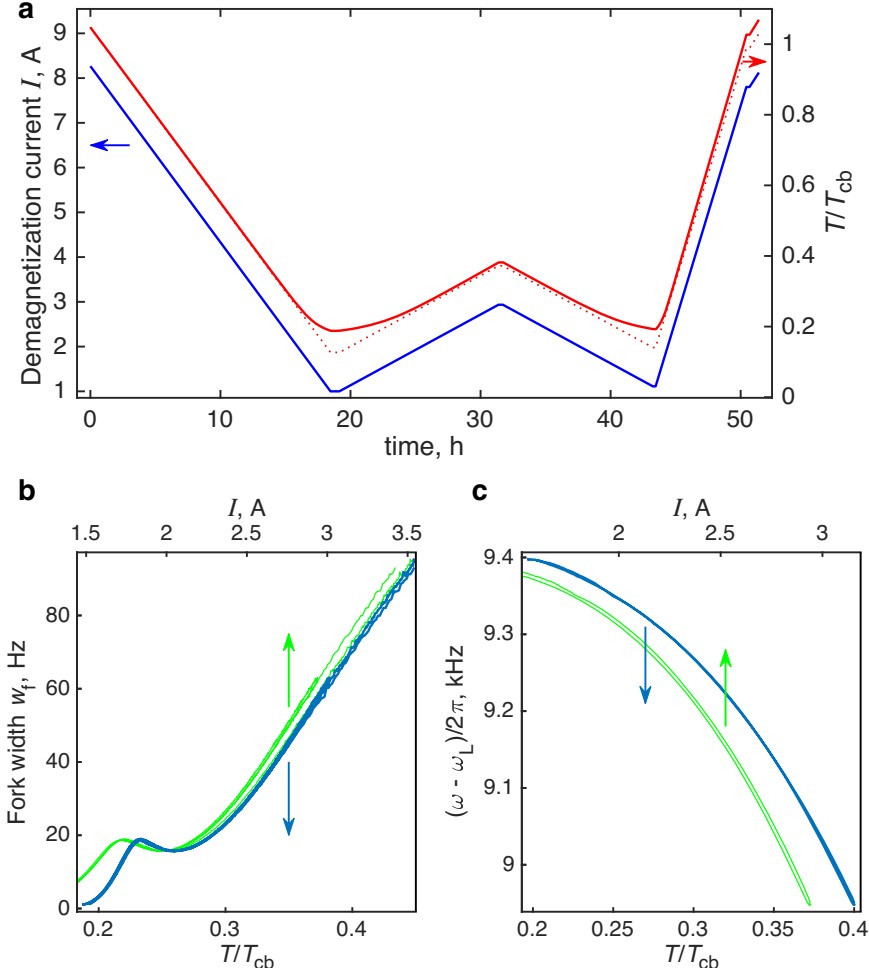

**Fig. 5 | An example of the measurement with the temperature calibration.**
**a** Dependence of the current $I(t)$ in the magnet of the demagnetization cooling stage on time $t$ for measurements at pressure of 11 bar (blue line, left $y$-axis), temperature $T^*(t)$ for ideal cooling in the absence of losses, heat leaks and thermal resistances (red dotted line, right $y$-axis), and the temperature of the sample $T(t)$, calculated using the thermal model (solid red line, right $y$-axis). **b** Resonance width $w_f$ of the quartz tuning fork installed in the bulk (B-phase) part of the sample as a

function of the demagnetization current $I$ (green line, upper $x$-axis) and as a function of the calculated sample temperature $T$ (blue line, lower $x$-axis). Lower $x$-axis also matches $T^*$ values corresponding to upper $x$-axis. The data measured in the time interval between 12 h and 46 h in the panel (**a**) are shown. **c** Similar plot as in panel (**b**) for the measured frequency shift $\omega(T) - \omega_L$ of the NMR response from the Larmor value. The data are collected in the time interval between 18.9 h and 43.3 h.

around the value of $H_L = \omega_{rf}/|\gamma| \approx 11$ mT to record the NMR spectrum. Here the Larmor field $H_L$ corresponds to location of the NMR response in normal $^3$He. In the polar phase, the NMR response shifts to lower fields $H < H_L$. For analysis, we convert the field shift to the equivalent frequency shift as $\omega - \omega_{rf} = \omega_{rf}(1 - H/H_L)$, which is applicable since the observed shifts are sufficiently small. We ensure that the magnetic field **H** is oriented along the nafen strands by rotating the field using 2-axis vector magnet and checking for the maximum value of the frequency shift.

The inhomogeneity of the applied magnetic field is $\Delta H/H \approx 10^{-4}$. This value determines the width of the NMR spectrum in the normal phase. In the polar phase when the temperature decreases the spectrum becomes wider due to inhomogeneity of the nafen, see Fig. 2a for spectra examples. To determine the frequency shift of the spectrum from the normal state to the lowest probed temperature, the first moment of absorption spectra are compared (the spectra are normalized to unit area). The spectrum shift with respect to the one measured at the lowest temperature is found by matching the whole absorption and dispersion profiles recorded at two temperatures using the relative shift as a fitting parameter. To accommodate the changing spectrum shape, the shift is determined relatively, within groups of 10 consecutively measured spectra, where the line shape

does not visually change. To verify that the error in accumulation of the relative shifts is not significant, we in parallel determine the NMR peak position by fitting a parabola close to the absorption maximum. Both methods produce the same overall dependence, but the peak-finding method results in significantly larger scatter, since it uses much fewer data from each recorded spectrum.

For pressures of 0.1, 4, 7, 11, and 15 bars one temperature sweep with increasing temperature and one with decreasing temperature are measured, while for 23 and 29.5 bars two pairs of increasing/decreasing temperature sweeps are recorded. All sweeps at the same pressure are averaged for the analysis.

**Thermometry**

Near the superfluid transition temperature, the NMR frequency shift of the bulk $^3$He (in the B or A phase, depending on pressure) is used as a thermometer. Small contribution to the NMR spectra from the unconfined helium is visible due to the geometry of the pick-up coils. At temperatures below about 0.7 $T_c$ the bulk signal is however too wide and too small to be distinguished from the noise. The experiment is equipped with a quartz tuning fork of nominal 32768 Hz frequency, which is immersed in bulk $^3$He. The width of the resonance $w_f$ of such fork is a sensitive probe of the temperature-dependent density of

quasiparticles and it is widely used as a thermometer in superfluid $^3$He measurements[61]. In our experiment, presence of the $^4$He film on all surfaces modifies the fork response, in particular with appearance of a pressure-dependent resonance-like feature seen in Fig. 5b. Thus, independent calibration of the fork becomes unreliable and challenging[62]. Nevertheless, the fork keeps its sensitivity in the whole studied temperature range, and we developed a procedure which allowed to recover the temperature assuming that the fork and NMR responses are reproducible for the same temperature.

The temperature sweep in the experiment is performed with the sweep of the current $I(t)$ in the demagnetization solenoid. For ideal cooling in the absence of losses, heat leaks and thermal resistances, the resulting temperature is $T^*(t) = T_{cb}[I(t)/I_{cb}]$. Here $I_{cb}$ is the demagnetization current at the moment of crossing $T_{cb}$ temperature as determined from the maximum resonance width of the tuning fork immersed in bulk liquid. To calculate the actual temperature of the sample $T(t)$ we use a thermal model which includes the nuclear stage and the sample cell with separate temperatures and the thermal resistance between them. The heat capacity of the stage and of the cell are assumed to be known (for the copper stage—from the design, for the cell—theoretically calculated from the known volume). The thermal resistance is taken as[60] $R_T \exp(\Delta/k_B T)$ where $\Delta$ is the value of the superfluid gap in the B phase, which is in the contact with the sinter, calculated with trivial strong-coupling corrections[29], and $R_T$ is an adjustable parameter. Other two parameters of the model are the heat leaks to the stage $q_{ns}$ and to the sample $q_{sam}$. We measure the NMR spectra and the fork resonance while sweeping the temperature up and down in the range of the interest with different rates. The model parameters are then selected in such a way, that at the same calculated temperature of the cell the measured shift of the NMR spectrum and the width of the fork resonance coincide independently of the direction and rate of the temperature sweep. An example of such temperature calibration is shown in Fig. 5. Same values of $q_{ns} = 2$ nW and $q_{sam} = 25$ pW are used at all pressures. The value of $R_T$ we characterize with the minimum attainable temperature of the sample $T_{min}$ as $R_T = T_{min} q_{sam}^{-1} \exp(-\Delta/k_B T_{min})$ and use $T_{min} = 0.17 T_{cb}$ at all pressures. These values agree with independent measurements of the heat leak to the nuclear stage and with expectations based on the previous experiments which had better thermal control of bulk $^3$He sample[63].

To estimate influence of possible temperature calibration error on the determination of the gap temperature dependence, we reanalyzed measurements at 7 bars with about 30 sets of the thermal model parameters distributed in the range of 0.5 to 2 times the values quoted above for the heat leaks and from 0.1 to 10 times for the thermal resistance. This range exceeds possible uncertainty in the parameters, as the fork and NMR responses become clearly hysteretic at the boundaries of the distribution. For all these sets, the fitted values of the gap power-law exponent are between 2.95 and 3.2 while the values of the parameter $a$ are between 0.47 and 0.58 when the fit is performed in the $(0.3-0.5)T_c$ temperature range.

## Data availability

The data generated in this study have been deposited in the Zenodo database under accession code https://doi.org/10.5281/zenodo. 8055891.

## Code availability

The code for the thermal model used in the temperature calibration is available from the corresponding author upon request.

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

## Acknowledgements

We thank Igor Fomin for discussions and Vladimir Dmitriev for providing nafen-243 sample. This work has been supported by the European Research Council (ERC) under the European Union's Horizon 2020 research and innovation programme (Grant Agreement No. 694248), by Academy of Finland (grant 332964), and additionally by the European Union's Horizon 2020 research and innovation programme under Grant Agreement No. 824109. The experiments were performed at the Low Temperature Laboratory, which is a part of the OtaNano research infrastructure of Aalto University and of the European Microkelvin Platform. T.K. acknowledges support from the Finnish Cultural foundation.

## Author contributions

The experiments were conducted by T.K., J.R., and M.-M.V. The data were analyzed by J.R. and V.B.E. Theoretical work was carried out by G.E.V. Writing was done by V.B.E. and G.E.V. with contributions from all the authors. V.B.E. supervised the project.

## Competing interests

The authors declare no competing interests.
