## [Peer Review File · Nature Communications]

REVIEWER COMMENTS

Reviewer #1 (Remarks to the Author):

The authors have carefully measured the temperature dependence of the superfluid gap in the polar phase of superfluid ^3He using NMR. The polar phase is stabilized by aligned strand like impurities in nafen aerogel, which act as dense impurities in the superfluid. Despite the presence of strong impurity scattering, the authors are able to see a T^3 dependence to the superfluid gap consistent with the line nodes predicted to be present in the polar phase. This observation, combined with relatively small T_c suppression and the relative agreement of the coefficient, a , of the T^3 dependence with calculations in the clean limit provide evidence for the extended Anderson theorem. However, as the researchers note, the impurities are not perfectly linear strands, and some small violation is expected due to pz scattering.

This is a unique experiment that exploits the purity of superfluid ^3He and the researcher's ability to add controlled impurity, something that is not possible in other superconducting systems. As the authors note, a demonstration of the extended Anderson theorem is relevant to other unconventional superconductors where control of disorder is not possible, and where control of gap suppression is necessary to make superconducting devices.

The experiment requires precise measurements of the NMR frequency shift and the authors have been careful in conducting their measurements and carrying out analysis. However, I have a few questions and suggestions listed below.

In general, I think the work is carefully carried out and relevant to a broader audience and would be useful to publish after some clarifications.

1) The authors interpretation hinges on accurately determining the power law of the gap temperature dependence. While the cubic fits in Figs. 2b and 3a look convincing, it would be useful to see the data plotted on a log-log plot to more clearly display the power law. Fitting the log-log dependence with the exponent α as a free parameter would provide more convincing evidence for how close the exponent is to 3 and allow the authors to provide error bars on the value.

2) Accurately measuring the temperature at low temperatures is challenging and the authors have carefully modeled the temperature, compared cooling and warming runs, and compared frequency shift and tuning fork measurements to calibrate their temperature. However, I would be interested to know how sensitive the power law and coefficient a are to the parameters used to calibrate the temperature.

3) The authors note that while a is close to the theoretical value, it is not exact and has unexpected pressure dependence. Some of which may come from strong coupling. Can the authors comment on whether this could be related to the fact that the aerogel strands are not perfect cylinders and therefore have some mixing of pz states leading to a small violation of Anderson's theorem?

4) The authors show increased line widths in the polar phase for the NMR spectra in Fig. 2a and note that they calculate frequency shift by shifting the spectra for comparison at different temperatures. Does the shape of the NMR spectra or its linewidth change significantly with temperature once it is in the polar phase?

Reviewer #2 (Remarks to the Author):

Recently, the polar p-wave pairing state has been realized experimentally in the liquid ^3He immersed in a couple of samples of nematic aerogel. Previously, identification of the new superfluid state in the aerogels with the equal-spin-paired polar state with a horizontal line node of the energy gap has been justified through the observation of the half-quantum vortices in the state. At this time, to obtain a firm evidence of the line gap-node, the authors have examined the NMR data to clarify the T^3

behavior of the quasiparticle energy gap at lower temperatures and have argued that the coefficient of the T^3 term takes a value close to the result in the BCS model in clean limit, and hence that the Anderson Theorem is nicely satisfied in the p-wave polar pairing state in the nematic aerogel used by them so that the topological line node is present with no impurity scattering effects by the aerogel structure.

Based on their data at $T > 0.2 T_c$, I think that their main conclusion that the line node structure of the energy gap is well defined in the polar pairing state realized in the random aerogel structure remains valid at least in such a temperature range. Looking at Fig.1d. however, I have to point out that some caution is needed in evaluating another conclusion that the Anderson Theorem is safely satisfied: Figure 1d implies that $T_c(P)$ deviates from the bulk T_c , $T_{cb}(P)$, in a manner dependent on the sample quality, implying that the Anderson Theorem is not satisfied for T_c . On the other hand, the authors argue that the low T power law of the energy gap is quantitatively consistent with the known result in the BCS model in clean limit if a sizable strong coupling correction is assumed to be effective even at lower pressures.

The apparent inconsistency between the T_c -deviation mentioned above and the well-defined T^3 behavior has been discussed in Ref.48 previously. By taking account of the fact that the strong coupling correction does not affect the mean field T_c , it is felt that the authors' data are consistently explained by incorporating both the impurity scattering-induced pressure dependence, indicated in Fig.3 as the two diamond symbols, and another pressure dependence due to the strong coupling correction.

In summary, I cannot recommend publication of the present work until a consistent explanation between the data of T_c and the energy gap is reached.

Reviewer #3 (Remarks to the Author):

The paper by Kappinen et al describes very interesting experimental results and interpretation of a limiting T^3 behavior at all pressures of the order parameter amplitude in terms of the equatorial nodal line expected for a pure polar phase. The interpretation is based on a clean BCS analysis that incorporates no impurities and specular boundary conditions making contact to the Anderson Theorem.

However, it is undeniable that there is a non-zero suppression of T_c that the authors measure, very similar to the work of Reference 22, i.e. larger at low pressure following what can be expected from the pressure dependence of the coherence length and some degree of dispersion of the scattering cross section perpendicular to the average strand direction and/or some degree of non-specularity. This gives rise to pair breaking, a subject addressed by Reagan et al. PRB 104, 024513 (2021) and especially their Eq. 7. The SEM picture in Fig. 1f, confirms this non-uniformity of the strand orientation, to some degree, just looking by eye. Consequently, there is evidence for the existence of pair-breaking and Andreev bound states. Bound states produce low lying quasiparticle states that modify the temperature dependence of the order parameter from what otherwise would be expected from the polar phase nodal line, especially in the low temperature limit. This temperature dependence would dominate T^3 .

On the other hand, the data give a credible presentation of T^3 behavior over the full pressure range despite difficulties with thermometry clearly stated in the paper by the authors. If this were not the case their experimental observations would be in keeping with their hypothesis based on an impurity-free polar phase despite the slight suppression of T_c and the claim that this is the low temperature limiting behavior. This is the major weakness of the paper.

We thank the reviewers for the careful evaluation of our manuscript and useful comments. Below we respond to the comments and describe changes to the manuscript.

Reviewer 1: The authors have carefully measured the temperature dependence of the superfluid gap in the polar phase of superfluid ^3He using NMR. The polar phase is stabilized by aligned strand like impurities in nafen aerogel, which act as dense impurities in the superfluid. Despite the presence of strong impurity scattering, the authors are able to see a T^3 dependence to the superfluid gap consistent with the line nodes predicted to be present in the polar phase. This observation, combined with relatively small T_c suppression and the relative agreement of the coefficient, a , of the T^3 dependence with calculations in the clean limit provide evidence for the extended Anderson theorem. However, as the researchers note, the impurities are not perfectly linear strands, and some small violation is expected due to p_z scattering.

This is a unique experiment that exploits the purity of superfluid ^3He and the researcher's ability to add controlled impurity, something that is not possible in other superconducting systems. As the authors note, a demonstration of the extended Anderson theorem is relevant to other unconventional superconductors where control of disorder is not possible, and where control of gap suppression is necessary to make superconducting devices.

The experiment requires precise measurements of the NMR frequency shift and the authors have been careful in conducting their measurements and carrying out analysis. However, I have a few questions and suggestions listed below.

In general, I think the work is carefully carried out and relevant to a broader audience and would be useful to publish after some clarifications.

We thank the reviewer for the positive evaluation of our manuscript.

1) The authors interpretation hinges on accurately determining the power law of the gap temperature dependence. While the cubic fits in Figs. 2b and 3a look convincing, it would be useful to see the data plotted on a log-log plot to more clearly display the power law. Fitting the log-log dependence with the exponent as a free parameter would provide more convincing evidence for how close the exponent is to 3 and allow the authors to provide error bars on the value.

The problem in determining the exponent is that data are not described by a pure power law, but by the power-law shift from the unknown zero-temperature value. In the log-log plot this zero-temperature value has to be subtracted, and depending on this hidden parameter, the plot may look very differently. In fact, the uncertainty in the zero-temperature value is the main contribution to the uncertainty in the exponent. Fixing the exponent to 3 allows very close fit to the data, and we decided to keep this approach in the main text of the manuscript. To address the reviewer's concerns, we added a new Supplementary Note 4, where we present log-log plots of all of the data with different zero-temperature offsets and different ranges for fit with the exponent in the power law used as a fit parameter (new Supplementary Figures 2 and 3).

To summarize this analysis, for pressures 0.1, 4, 7 and 29 bars, the interval of the possible exponent values can be determined as 2.9 – 3.15. Pressures 11, 15 and 23 bars are different, with the most prominent anomalous behavior observed at 15 bars. With more analysis performed, we now believe that there is a physical reason for this behavior. When all the temperatures below $0.5T_c$ are included in the fit, the exponent for these three pressures reaches values of 3.7 – 4. When the fit range is limited to higher temperature end, the exponent decreases. We interpret this observation as an indication of the phase transition at the lowest temperatures at these pressures with the most likely candidate for the new phase being the polar-distorted A (PdA) phase. This is described in more details in the added Supplementary Note 4 and the text of the manuscript is updated accordingly with the reference to the phase transition.

To avoid the effect of the suspected phase transition at elevated pressures on the analysis in the main text, we shifted the fit range for that analysis (with the fixed exponent 3) from $(0.2 - 0.4)T_c$ in the original submission to $(0.3 - 0.5)T_c$ in the revised version (for all pressures for consistency). We concluded that $0.5T_c$ is a safe upper temperature for the cubic fit based on the simulation results in Fig. 4 of the Hisamitsu et al paper [Ref. 38 of the manuscript]. The change of the fit range affected visually results only at 11, 15 and 23 bars, which we believe represent now better the polar phase.

We stress that all messages of the manuscript are proved by the data at 0.1, 4 and 7 bar alone without complications of the phase transition, see points at those pressures in Supplementary Figure 2. We,

however, believe that we have found a reasonable way to extract the polar-phase behavior from the higher-pressure data as well, which gives access to a wider range of parameters.

2) Accurately measuring the temperature at low temperatures is challenging and the authors have carefully modeled the temperature, compared cooling and warming runs, and compared frequency shift and tuning fork measurements to calibrate their temperature. However, I would be interested to know how sensitive the power law and coefficient a are to the parameters used to calibrate the temperature.

These parameters are three: heat leak to the nuclear stage, heat leak to the sample and the prefactor in the exponentially growing thermal resistance between the nuclear stage and the sample. In fact, these parameters are known with some certainty from independent measurements. Heat leak to the nuclear stage was measured multiple times over years with a stable value slightly below 2 nW found and for the heat leak to the sample and for the thermal resistance we have estimations from earlier experiments which had better thermal control of bulk ^3He . The used parameters of the model are close to the expectations. To answer the reviewer's question, we reanalyzed the measurement at 7 bars with many sets of the thermal model parameters distributed in the range of 0.5 to 2 times the original value for the heat leaks and from 0.1 to 10 times for the thermal resistance. This range is certainly larger than possible uncertainty in the parameters, as the fork and NMR responses become clearly hysteretic at the boundaries of the distribution. For all these sets, the fitted values of the exponent are between 2.95 and 3.2 while the values of the parameter a are between 0.47 and 0.58.

The shift of the temperature range used for fitting to higher temperatures in the revised version reduced dependence on the thermal model parameters since the temperature correction is more important at the lower temperatures (difference between solid and dotted red lines in Fig. 5a). Also, dependence on the thermal model parameters is not very strong because we always measure up and down temperature sweeps and fit the average value. Even when hysteresis is developed due to inaccurate temperature determination, the averaging of the two sweep directions partially compensates this error.

The thermometry subsection of the Methods has been extended and now includes discussion on the sensitivity to parameters of the thermal model.

3) The authors note that while a is close to the theoretical value, it is not exact and has unexpected pressure dependence. Some of which may come from strong coupling. Can the authors comment on whether this could be related to the fact that the aerogel strands are not perfect cylinders and therefore have some mixing of pz states leading to a small violation of Anderson's theorem?

This was a common suggestion between the reviewers. The short answer is yes. The detailed response is given in connection with comments of the Reviewer 2.

4) The authors show increased line widths in the polar phase for the NMR spectra in Fig. 2a and note that they calculate frequency shift by shifting the spectra for comparison at different temperatures. Does the shape of the NMR spectra or its linewidth change significantly with temperature once it is in the polar phase?

Yes, the line shape continues to change also in the polar phase and the pattern of the change versus pressure and temperature may be rather complicated. The origin is not understood yet, but probably is related to inhomogeneity of the sample. To alleviate this problem in the analysis, the shift between spectra is determined relatively, within groups of 10 spectra, where the line shape does not visually change. To check that there is no mistake in the analysis and that the error in accumulation of the relative shifts is not significant, we in parallel determine the NMR peak position in a traditional way, by fitting parabola close to the maximum. The traditional method gives the same result, but significantly more scattered, which is quite expected, since it uses just a few points in the absorption channel from a whole spectrum. On the contrary, our method utilizes many points from both absorption and dispersion channels. It is explained now in more details in the Methods section.

Reviewer 2: Recently, the polar p-wave pairing state has been realized experimentally in the liquid ^3He immersed in a couple of samples of nematic aerogel. Previously, identification of the new superfluid state in the aerogels with the equal-spin-paired polar state with a horizontal line node of the energy gap has been justified through the observation of the half-quantum vortices in the state. At this time, to obtain a firm evidence of the line gap-node, the authors have examined the NMR data to clarify the T^3 behavior of the quasiparticle energy gap at lower temperatures and have argued that the coefficient of the T^3 term takes a value close to the result in the BCS model in clean limit, and hence that the Anderson Theorem is nicely

satisfied in the p -wave polar pairing state in the nematic aerogel used by them so that the topological line node is present with no impurity scattering effects by the aerogel structure.

Based on their data at $T > 0.2T_c$, I think that their main conclusion that the line node structure of the energy gap is well defined in the polar pairing state realized in the random aerogel structure remains valid at least in such a temperature range. Looking at Fig.1d. however, I have to point out that some caution is needed in evaluating another conclusion that the Anderson Theorem is safely satisfied: Figure 1d implies that $T_c(P)$ deviates from the bulk T_c , $T_{cb}(P)$, in a manner dependent on the sample quality, implying that the Anderson Theorem is not satisfied for T_c . On the other hand, the authors argue that the low T power law of the energy gap is quantitatively consistent with the known result in the BCS model in clean limit if a sizable strong coupling correction is assumed to be effective even at lower pressures.

We are happy that the Reviewer agrees with the conclusion on the nodal line in the polar phase. With respect to the Anderson theorem, we point out that the original version of the manuscript already noted that in the real nafen material there are mechanisms which do not conserve p_z on the scattering from the strands and thus some violations are expected and observed. We tried to make this more clear in the revised manuscript. We stress, however, that a remarkable finding is that although scattering is strong and not fully ideal, the observed behavior is nearly ideal clean-limit behavior.

The deviations from the ideal behavior are observed both at T_c and at low temperatures. The common question of the Reviewers is whether these deviations are linked together. This is an important question and it was indeed overlooked in the original version of the manuscript. We checked more carefully our data and found a confirmation that this link may exist. On Fig. 3b in the updated version, we additionally plot T_c suppression as a function of pressure. When the temperature axis is appropriately scaled, one finds that the pressure dependence of T_c and that of a are similar. Thus, one possible source for the pressure dependence of a may have the same origin as the T_c suppression, that is, violation of the Anderson theorem due to non-ideal scattering. The discussion of this observation was added to the manuscript. Another possible contribution to the pressure dependence of a is discussed below the next remark of the Reviewer.

We also added to the manuscript a discussion on what are the expectation of the role of disorder in the nodal-line systems if the Anderson theorem is not working. In particular, we compared our measurements with the model of Ref. 25 which predicts change of the power law to 2.14. Our observations are clearly inconsistent with this prediction.

The apparent inconsistency between the T_c -deviation mentioned above and the well-defined T^3 behavior has been discussed in Ref.48 previously. By taking account of the fact that the strong coupling correction does not affect the mean field T_c , it is felt that the authors' data are consistently explained by incorporating both the impurity scattering-induced pressure dependence, indicated in Fig.3 as the two diamond symbols, and another pressure dependence due to the strong coupling correction. In summary, I cannot recommend publication of the present work until a consistent explanation between the data of T_c and the energy gap is reached.

The Reviewer is correct that the model which allows to link T_c suppression and the low-temperature gap dependence was suggested in former Ref. 48 (now Ref. 38) of the manuscript. The model correctly predicts the survival of the T^3 behavior and the decrease of the value of a with decrease of T_c suppression as pressure increases. We now acknowledge this agreement in the manuscript. We have fixed a typo in the manual typing of Ref. 38 data to the plotting script ($0.81 \rightarrow 0.71$), which improved agreement at low pressures even further. One can conclude from this comparison that upward increase of a at the lowest pressure from the clean-limit weak-coupling value is due to (weak) violation of the Anderson theorem in real confining material. The pressure dependence of a from Ref. 38 while qualitatively correct does not match the observations.

We added the strong-coupling rescaling of $\Delta(0)/T_c$ to Ref. 38 calculations as the Reviewer suggested. Accepted values from the B phase are not sufficient to explain the difference to the measurements (filled triangle in the updated Fig. 3b). We also performed new analysis, using zero-temperature frequency shift to find strong-coupling corrections in the polar phase (new Supplementary Fig. 4). With this simple approach, a better match to the measurements is obtained (empty triangle in Fig. 3b for high-pressure Ref. 38 data). However, it has limitations mentioned in the manuscript.

Overall, it is likely that residual violation of the Anderson theorem by the real-material disorder and strong-coupling effects are both responsible for the observed deviations of the low-temperature behavior

of the gap in the polar phase from the clean-limit weak-coupling predictions. We believe that the detailed theoretical description of these effects, including possible contribution of the bound states mentioned by the Reviewer 3, remains a task for future, while our manuscript concentrates on the main phenomenon: Experimental demonstration of the extension of the Anderson theorem to the topological p -wave system.

Reviewer 3: The paper by Kappinen et al describes very interesting experimental results and interpretation of a limiting T^3 behavior at all pressures of the order parameter amplitude in terms of the equatorial nodal line expected for a pure polar phase. The interpretation is based on a clean BCS analysis that incorporates no impurities and specular boundary conditions making contact to the Anderson Theorem.

We are happy that the Reviewer finds our work very interesting.

However, it is undeniable that there is a non-zero suppression of T_c that the authors measure, very similar to the work of Reference 22, i.e. larger at low pressure following what can be expected from the pressure dependence of the coherence length and some degree of dispersion of the scattering cross section perpendicular to the average strand direction and/or some degree of non-specularity. This gives rise to pair breaking, a subject addressed by Reagan et al. PRB 104, 024513 (2021) and especially their Eq. 7. The SEM picture in Fig. 1f, confirms this non-uniformity of the strand orientation, to some degree, just looking by eye. Consequently, there is evidence for the existence of pair-breaking and Andreev bound states.

As described above, the discussion of violations of the Anderson theorem is significantly enhanced in the manuscript. We also added the suggested reference, and used Eq. 7 from it to fit the transition temperatures in Fig. 1d (instead of guides to the eye shown in the previous version). The fits to the Dmitriev's and our data are performed with the same density (and radius) of the strands, since nominally the same material is used, but with different scattering radius along the strands (σ_{\parallel}). The reasonable fit is achieved which confirms the claim of the manuscript that it is not the defect density, which controls T_c suppression (in accordance with the Anderson theorem), but deviation of the scattering properties from ideal (e.g., orientational disorder of the strands).

Bound states produce low lying quasiparticle states that modify the temperature dependence of the order parameter from what otherwise would be expected from the polar phase nodal line, especially in the low temperature limit. This temperature dependence would dominate T^3 .

It is true that the bound states may dominate behavior at the lowest temperatures. The question is at which temperature it might be expected. Measurements of the heat capacity of $^3\text{He-B}$ [Bunkov & Gazizulin, Sci. Rep. **10**, 20120 (2020)] show that the heat capacity of the bound states becomes noticeable only below $0.2T_c$ and dominates over bulk below $0.15T_c$. And this is found in the gapped superfluid, where contribution from bulk freezes out exponentially with decreasing temperature. One may expect that in the polar phase, where the bulk contribution decreases only as a power law, this crossover happens at even lower temperature, despite larger surface area. Nevertheless, we have increased in the revised version the lower temperature bound for T^3 fit from $0.2T_c$ to $0.3T_c$ also to avoid the problem with the bound states.

On the other hand, the data give a credible presentation of T^3 behavior over the full pressure range despite difficulties with thermometry clearly stated in the paper by the authors. If this were not the case their experimental observations would be in keeping with their hypothesis based on an impurity-free polar phase despite the slight suppression of T_c and the claim that this is the low temperature limiting behavior. This is the major weakness of the paper.

We are happy that the Reviewer finds T^3 observations credible. We disagree with the evaluation that the thermometry is the major weakness of the paper. That would have been a valid point if a known alternative method would exist, which we neglected to implement. In fact, there is no such method known to us. Traditionally, for measurements of ultra-low temperatures in superfluid ^3He , immersed mechanical resonators are used with great success. However, a recent work (Ref. 62) showed that in the case of ^4He surface preplating, needed to stabilize the polar phase, the behavior of mechanical resonators changes and becomes unsuitable for direct temperature measurements. Thus our method is actually a significant experimental advance, which enabled discovery of T^3 law. The exaggerated uncertainty in the exponent following from possible temperature measurement errors is quoted in the response to the Reviewer 1.

As additional changes, we reformatted the manuscript to the Nature Communication requirements; moved

two extended data figures to the main text and two others to supplementary information; moved some information from figure captions to the text, to reduce the captions size; fixed a typo in the lowest temperature in the Sample subsection of Methods.

Vladimir Eltsov
(on behalf of all authors)

REVIEWERS' COMMENTS

Reviewer #1 (Remarks to the Author):

The resubmitted manuscript has been improved by the authors. The authors have added more discussion and analysis on their fitting procedures, temperature calibration, and frequency shift determination which answer my questions and show that the authors have carefully carried out the measurements and analysis to show the T^3 dependence of the superfluid gap and determine the coefficient of this term.

All of the reviewers pointed out that the slight deviations of the aerogel strands from perfectly aligned cylinders provide weak violation of Anderson's theorem and may be responsible for deviations from the expected clean limit seen in the experiment. The authors have responded by making this violation more explicit in the text and comparing their data to several theories that include small amounts of pz scattering. They have also demonstrated that the suppression of T_c and the energy gap behave similarly as expected if the deviations come from small pz scattering. While this small scattering violates the perfect realization of Anderson's theorem, demonstrating only small deviations of T_c and the gap from clean limit values, despite much larger disorder density than is present in previous studies in silica aerogel is an experimental achievement and provides strong evidence for Anderson's theorem.

I believe that the improved manuscript is clear on this point and worthy of publication.

Reviewer #2 (Remarks to the Author):

In the second (revised) manuscript, the authors seem to have nicely responded to requirements and recommendations given in my first report. An important point in the present manuscript is that the smallness, particularly at lower pressures, of $|\Delta|$ has been repeatedly stressed in the text together with the absence of superfluid phases other than the polar one and a small reduction of T_c even at lower pressures. Due to this, the argument favoring the Anderson Theorem in spite of the fact that the measurements have been limited to those at higher temperatures than $0.3 T_c$ is sufficiently convincing. I will recommend publication of the present work.

Reviewer #3 (Remarks to the Author):

The resubmission of the paper Topological Nodal Line in Superfluid ^3He and the Anderson Theorem by Kamppinen et al., accompanied by a detailed explanation of changes in the manuscript and a reasonable modification of the claims, is a serious work and, in my view, satisfies the requirements for publication.

We thank the reviewers for the positive evaluation of our manuscript. Since no comments which require changes to the manuscript were made, below we simply reproduce the reviewers comments.

Reviewer 1: The resubmitted manuscript has been improved by the authors. The authors have added more discussion and analysis on their fitting procedures, temperature calibration, and frequency shift determination which answer my questions and show that the authors have carefully carried out the measurements and analysis to show the T^3 dependence of the superfluid gap and determine the coefficient of this term.

All of the reviewers pointed out that the slight deviations of the aerogel strands from perfectly aligned cylinders provide weak violation of Anderson's theorem and may be responsible for deviations from the expected clean limit seen in the experiment. The authors have responded by making this violation more explicit in the text and comparing their data to several theories that include small amounts of pz scattering. They have also demonstrated that the suppression of T_c and the energy gap behave similarly as expected if the deviations come from small pz scattering. While this small scattering violates the perfect realization of Anderson's theorem, demonstrating only small deviations of T_c and the gap from clean limit values, despite much larger disorder density than is present in previous studies in silica aerogel is an experimental achievement and provides strong evidence for Anderson's theorem.

I believe that the improved manuscript is clear on this point and worthy of publication.

Reviewer 2: In the second (revised) manuscript, the authors seem to have nicely responded to requirements and recommendations given in my first report. An important point in the present manuscript is that the smallness, particularly at lower pressures, of $\tau|\Delta|$ has been repeatedly stressed in the text together with the absence of superfluid phases other than the polar one and a small reduction of T_c even at lower pressures. Due to this, the argument favoring the Anderson Theorem in spite of the fact that the measurements have been limited to those at higher temperatures than $0.3T_c$ is sufficiently convincing. I will recommend publication of the present work.

Reviewer 3: The resubmission of the paper Topological Nodal Line in Superfluid ^3He and the Anderson Theorem by Kamppinen et al., accompanied by a detailed explanation of changes in the manuscript and a reasonable modification of the claims, is a serious work and, in my view, satisfies the requirements for publication.

We are thankful to the reviewers for the useful comments which allowed us to improve our manuscript.

Vladimir Eltsov
(on behalf of all authors)